# Comparative proteomic analysis of silica-induced pulmonary fibrosis in rats based on tandem mass tag (TMT) quantitation technology

**Cunxiang Bo**[1][⊙], **Xiao Geng**[1][⊙], **Juan Zhang**[1], **Linlin Sai**[1], **Yu Zhang**[1], **Gongchang Yu**[1], **Zhenling Zhang**[1], **Kai Liu**[2], **Zhongjun Du**[1], **Cheng Peng**[3], **Qiang Jia**[1]*, **Hua Shao**[1]*

1 Shandong Academy of Occupational Health and Occupational Medicine, Shandong First Medical University & Shandong Academy of Medical Sciences, Ji'nan, Shandong, China, 2 Department of Cardiovascular Surgery, Qilu Hospital of Shandong University, Ji'nan, Shandong, China, 3 Queensland Alliance for Environmental Health Sciences, The University of Queensland, Brisbane, Queensland, Australia

⊙ These authors contributed equally to this work.
* jiaqiang5632@163.com (QJ); chinashaohua5888@163.com (HS)

**Data Availability Statement:** The mass spectrometry proteomics data have been deposited to the ProteomeXchange Consortium via the iProX partner repository with the dataset identifier PXD020625.

## Abstract

Silicosis is a systemic disease characterized by chronic persistent inflammation and incurable pulmonary fibrosis with the underlying molecular mechanisms to be fully elucidated. In this study, we employed tandem mass tag (TMT) based on quantitative proteomics technology to detect differentially expressed proteins (DEPs) in lung tissues of silica-exposed rats. A total of 285 DEPs (145 upregulated and 140 downregulated) were identified. Gene Ontology (GO) and Kyoto Encyclopedia of Genes and Genomes (KEGG) analyses were performed to predict the biological pathway and functional classification of the proteins. Results showed that these DEPs were mainly enriched in the phagosome, lysosome function, complement and the coagulation cascade, glutathione metabolism, focal adhesion and ECM-receptor interactions. To validate the proteomics data, we selected and analyzed the expression trends of six proteins including CD14, PSAP, GM2A, COL1A1, ITGA8 and CLDN5 using parallel reaction monitoring (PRM). The consistent result between PRM and TMT indicated the reliability of our proteomic data. These findings will help to reveal the pathogenesis of silicosis and provide potential therapeutic targets. Data are available via ProteomeXchange with identifier PXD020625.

## Introduction

Silicosis represents a pulmonary fibrosis disease caused by long-term inhalation of free silica dust, primarily produced in the mining and construction industries [1]. It is the most serious occupational lung disease especially in developing countries [2, 3], affecting the quality of life of individuals. The main characteristic histopathological feature of silicosis is the build-up and accumulation of fibrosing nodular lesions with progressive massive fibrosis and gradual loss of respiratory functions [4, 5]. The disease is characterized by fatal, irreversible, incurable signs

**Funding:** This study was supported by Natural Science Foundation of Shandong (ZR2017YL001), the Innovation Project of Shandong Academy of Medical Sciences, Academic promotion programme of Shandong First Medical University (2019QL001), the Department of Science and Technology of Shandong Province (2018GSF118212, 2018GSF121007), China Coal Miner Pneumoconiosis Prevention Treatment Foundation (201915J039), the National Nature Science Foundation of Chian (No.81872603, 81600293). The funders had no role in study design, data collection and analysis, decision to publish, or preparation of the manuscript.

**Competing interests:** Our research is supported only by government fundings. The authors have read the journal's policy and declare that they have no known competing financial interests or personal relationships that could have appeared to influence the work reported in this paper.

and fibrosis develops even if the exposure is terminated [6, 7]. The progression of pulmonary fibrosis in patients can't be halted or reversed after diagnosis for lack of effective treatment [8]. At present, the exact pathogenesis of silicosis is still unclear and there is no effective early diagnosis method and health monitoring biomarkers for silicosis patients and exposed workers. Therefore, exploration of the pathogenesis and potential biomarkers for early diagnosis of silicosis represent an urgent issue to be solved.

The mechanism of occurrence and development of silicosis is related to the abnormal change of various proteins. Comparative proteome research is used to reveal the protein regulatory network in the process of disease occurrence and find the key or new drug target proteins. Previous proteome research in silicosis is based on two-dimensional gel electrophoresis (2 DE) and matrix assisted laser desorption ionization (MALDI) time of flight (TOF)-mass spectrometer (MS) analysis [9, 10]. However, this approach is not ideal for its lack of sensitivity and accuracy. In addition, the experimental processes are time-consuming and laborious with difficulties to analyze smaller or larger molecular weight proteins, low abundance proteins, and extremely alkaline and hydrophobic proteins. Isotope-labeling measuring techniques (isobaric tags for relative and absolute quantitation/Tandem Mass Tag, iTRAQ/ TMT) improve the accurate and relative quantification of proteins identification, which is one of the most sensitive techniques currently used in comparative proteomics [11, 12]. It has the potential to reveal novel diagnostic and therapeutic targets as well as potential biomarkers [13]. Some studies have been done on the pathogenesis of silicosis in vitro by iTRAQ-coupled 2D LC-MS/MS [14, 15]. At present, iTRAQ/TMT has not been used to study the proteome of lung tissue in silicosis model rats.

In this study, we examined and analyzed the differentially expressed proteins (DEPs) in lung tissues of silica-exposed rats by TMT combined with liquid chromatography-mass spectrometry (LC-MS/MS) to gain a wide and complete understanding of the protein regulatory network in the process of silicosis. Furthermore, parallel reaction monitoring (PRM) was applied to further verify the results of TMT. We investigated the key proteins related to silicosis and provided new targets for the origin and development as well as diagnosis, prevention and treatment of silicosis. We expect that this dataset will provide the foundation for further mining of disease-specific biomarkers for silicosis and implementing early intervention.

## Materials and methods

### Ethics statement

All experiments related to care and use of rats were performed in accordance with the National Institutes of Health guidelines for care and use of animals. In addition, these experiments were also approved by the Committee on the Ethics of Animal Experiments of Shandong Academy of Occupational Health and Occupational Medicine (Protocol Number: 20190003). All rats were intratracheally instilled with silicon dioxide under sodium pentobarbital anesthesia and sacrificed by carbon dioxide anesthesia after the exposure. All efforts were made to minimize suffering.

### Animals and treatments

Twenty specific-pathogen-free (SPF) male *Wistar rats*, 6–8 weeks in age, were purchased from Beijing Vital River Laboratory Animal Technology Co., Ltd. (Beijing, China) and housed in an SPF facility (temperature 20–24˚C; relative humidity 50–55%; light-dark cycle 12/12 h) with free access to food and water. After one week, the rats were randomly divided into two groups: model group (n = 10) intratracheally instilled with 1.0 mL (50 mg/mL) silica suspension (silica particles, #BCBW4148, Sigma Aldrich, USA) as previously described [16]; control group

(n = 10) intratracheally instilled with 1.0 mL normal saline. All rats were sacrificed on day 28 after treatment.

## Histopathologic examination

The right lower lungs from all rats were isolated and fixed with 4% paraformaldehyde for 24 hours, dehydrated in a series of graded ethanol solutions, then embedded in paraffin. Serial paraffin sections were cut in 4 μm thick. Subsequently, sections were stained with hematoxylin-eosin (HE) and masson trichrome to evaluate the histopathological changes in lung. The score of alveolitis and pulmonary fibrosis was determined as previously described [17]. The left lungs were preserved immediately snap-frozen in liquid nitrogen and stored at −80˚C.

## Extraction of total protein from lung tissue

Based on the results of the histological examination, three lungs were selected from each group for quantitative proteomic analysis. The procedures for protein preparations were according to previous papers [18]. Briefly, 300 μL lysis buffer SDS (P0013G, Beyotime Biotechnology, China) and 1 mM protease inhibitor Phenylmethanesulfonyl fluoride (PMSF, PB0425-5G, Amresco, USA) were added to the frozen sample, followed by ultrasonic treatment (1 s/1 s intervals, 80 W) on ice for 3 min and centrifugation (12,000 $g$, 4˚C) for 10 min. The supernatant was collected, packaged and frozen at −80˚C. The protein concentrations were assayed by the method of BCA (23227, Thermo, USA) according to the manufacturer's instructions. Next, 12% SDS-PAGE (17-1313-01, Sinopharm, China) was applied to separate 10 μg protein from each sample. The corresponding protein bands were observed by Coomassie Blue R-250 staining to conform the quality of proteins.

## Protein digestion and TMT labeling

Protein digestion was carried out as previously described [19]. After protein quantification, 100 μg of protein was incubated with 120 μL reduction buffer 10 mM DL-Dithiothreitol (DTT, A620058-0005, Sangon Biotech, China), 8 M Urea, 100 mM triethyl-ammonium bicarbonate (TEAB, A510932-0500, Sangon Biotech, China, pH 8.0) at 60˚C for 1 h. Next, add indole-3-acetic acid (IAA, A600539-0005, Sangon Biotech, China) to a final concentration of 50 mM at room temperature in the dark for 40 min, followed by centrifugation (12,000 $g$, 4˚C) for 20 min. 100 μL in 300 mM TEAB was added and centrifuged for three time. At last time 2 μL trypsin (HLSTRY001C, HualishiTechnology Co., Ltd., China) in 1 μg/μL was added and incubated at 37˚C for 12 h. Finally, 50 μL in 100 mM TEAB were added and centrifuged again. The digested peptides were collected and solubilized using 100 μL in 200 mM TEAB and 40 μL of each sample was labelled. TMT Reagents (TMT6 Label Reagents, 90066B, Thermo Scientific) were carried out according to the manufacturer's instructions. Briefly, 41 μL of the TMT label reagent was added to each sample carefully followed by incubation at room temperature for 1 h while mixing. The reaction was quenched with 8 μL of 5% (w/v) hydroxylamine in TEAB. Samples were pooled together and stored at −80˚C prior to LC−MS/MS analysis. Proteomics platform was provided by Shanghai luming biological technology co., LTD (Shanghai, China).

## LC-MS/MS analysis

LC-MS/MS was performed using a Q Exactive mass spectrometer (Thermo Scientific) combined with Easy nLC system 1200 (Thermo Scientific). Peptides were trapped on Agilent 1100 HPLC System with a C18 column (5 μm, 150 mm × 2.1 mm, C18, Agilent Zorbax Extend, USA) and separated on a C18-reversed phase analytical column (150 mm ×75 μm, 2 μm, 100 A, Acclaim PepMap RSLC, USA). The analytical separation was run for 90 min using agradient

of solution A (formic acid, concentration 0.1%) and solution B (acetonitrile 80% and formic acid 0.1%). The multistep gradient: 0~55 min, 8% B; 55~79 min, 30% B; 79~80 min, 50% B; 80~90 min, 100% B. Full MS scans were acquired in the mass range of 300–1600 m/z with a mass resolution of 70000 and the AGC target value was set at 1e6. The ten most intense peaks in MS were fragmented with higher-energy collisional dissociation (HCD) with NCE of 32. MS/MS spectra were obtained with a resolution of 17500 with an AGC target of 2e5 and a max injection time of 80 ms. The Q-E dynamic exclusion was set for 15.0 s and run under positive mode.

## Mass spectrometry data and bioinformatic analysis

Proteome Discoverer (v.2.2, Thermo, America) was used to search all of the Q Exactive raw data thoroughly against the UniProt database (https://www.uniprot.org/). Various search parameters were set: a peptide mass tolerance of ±10 ppm, variable modifcations of oxidation (M), a fragment mass tolerance of 0.02 Da, decoy as the database pattern and a peptide false discovery rate (FDR) of ≤0.01. For protein quantization, the protein was required to contain at least one unique peptide. The quantitative protein ratios were weighted and normalized by the median ratio in Mascot [20, 21]. For three biological replicates, the ratio of mean expression between model and control was defined as fold changes (FC). Those proteins with significant differences between control and model groups are considered DEPs.

The DEPs were performed using the Database for Annotation Visualization and Integrated Discovery (DAVID). The biological process (BP), cellular component (CC) and molecular function (MF) were annotated by the Gene Ontology (GO) database. The signaling pathways of proteins were elucidated by searching against the Kyoto Encyclopedia of Genes and Genomes (KEGG) database. The protein-protein interaction (PPI) of selected proteins was analyzed by Search Tool for the Retrieval of interacting Genes/proteins (STRING) Software.

## Verification of protein expression levels by PRM

The candidate proteins were verified by PRM on a Q Exactive mass spectrometer (Thermo Scientific) combined with Easy-nLC system1200 (Thermo Scientific). The lung tissues used for RPM validation were same to the TMT analysis and the peptides were prepared according to TMT. Tryptic peptides of each sample were spiked with the equivalent heavy isotope AQUA peptide (an internal standard) [22] and loaded onto a C18 column (75 μm × 15 cm, C18, 3 μm, 120 A, hromXP Eksigent, America). The full MS scan from 350 to 1650 m/z was acquired with an orbitrap resolution of 30000 (at m/z 200), AGC value was set to 3e6 and 200 ms maximum ion injection time (IT). Ion activation and dissociation was performed at normalized a collision energy of 27 in HCD collision cells. Following this step, 20 MS2 scans (target precursor ions) were performed and orbitrap resolution was set to 30000 (at m/z 200), isolation window was set to 1.6Th. ACG target value was set to 3e6 and maximum IT was set to 120 ms. The raw data were analyzed using Skyline (MacCoss Lab, University of Washington) software (V.4.2).

## Data sharing

The mass spectrometry proteomics data have been deposited to the ProteomeXchange Consortium (http://proteomecentral.proteomexchange.org) via the iProX partner repository [23] with the dataset identifier PXD020625.

## Statistical analysis

Statistical analysis was performed with Statistical Program for Social Sciences (SPSS) (SPSS Inc., version 20.0, United States). The quantitative data were reported as the means ± Standard

Deviation (SD), and the significant difference was analyzed with t-test between two groups, P values <0.05 was considered statistically significant. In TMT proteins with p values <0.05 and fold changes ≥±1.8 were considered as DEPs. A multiple testing correction was performed using Benjamini and Hochberg procedure to control the False Discovery Rate (FDR), using P value (<5%) [24]. GO and KEGG analyses were carried out using Fisher's exact test, using the entire quantifed protein annotations as the background dataset. Only categories and pathways with p values <0.05 were considered statistically significant.

## Results

### Histopathological evaluation of lung tissue

Lung architecture was normal in sections of lungs from the control group (Fig 1A and 1C). HE staining showed thickened alveolar walls, damaged alveolar structures, more infiltrating inflammatory cells and silicotic nodules (Fig 1B), and Masson staining showed damaged alveolar septa, diffuse silicon nodules, more collagen fibers and inflammatory cell infiltration in model group (Fig 1D). Lungs of model group showed a marked increase in scores of alveolitis and pulmonary fibrosis compared with the control group (P<0.05) (Table 1).

### Protein identification and differential expression

A total of 3099 proteins were identified (S1 and S2 Tables), of which 285 DEPs (145 upregulated and 140 downregulated) were identified between the control and model groups respectively (Fig 2A, S3 and S4 Tables). Heat maps were generated using these 285 DEPs (Fig 2B). The relative expression levels are shown by the intensity of the color. Red, green, or black colors indicate relative increase, decrease, or no quantitative information regarding protein content for a particular protein.

### PRM validation of the protein expression

Six significantly changed proteins, including CD14 (UniProt identifier Q63691), PSAP (UniProt identifier P10960), GM2A (UniProt identifier Q6IN37), COL1A1 (UniProt identifier P02454), CLDN5 (UniProt identifier Q9JKD6) and ITAG8 (UniProt identifier B5DEG1) were examined by PRM. These proteins have larger FC value and potentially important biological functions related to inflammation or fibrosis. The results showed that the expression levels of CD14、PSAP、COL1A1 and GM2A were all increased, and ITAG8 and CLDN5 were decreased in the model group than those in the control group. This was exactly the same trend as that observed when the protein levels were quantified by TMT (Fig 3).

### Bioinformatic analysis

Using DAVID software to initially explore the potential functions of those DEPs in silicosis. The colors of the bar charts represent the top ten terms of the three different categories (Fig 4). For BP, response to external stimulus, immune system process and cellular response to chemical stimulus were the top three significantly enriched terms (blue in Fig 4). For CC, extracellular region part, extracellular region and extracellular vesicle were the top three significantly enriched terms (red in Fig 4). For MF, protein binding, cell adhesion molecule binding and lipid binding were the top three significantly enriched terms (yellow in Fig 4).

Data from KEGG pathway analysis indicated that DEPs between the groups were involved in 34 pathways, Fig 5 shows the top twenty terms. The results showed that the DEPs were enriched in phagosome (14 proteins), lysosome (8 proteins), leukocyte transendothelial migration (8 proteins), cell adhesion molecules (CAMs) (8 proteins) and focal adhesion (7 proteins),

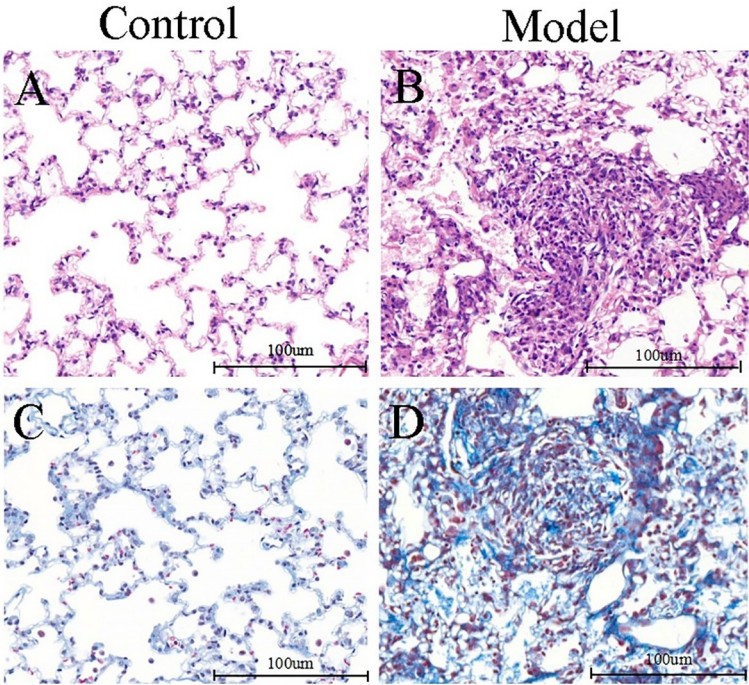

**Fig 1. Histological examinations of lung of rat from the control and model groups.** A: Lung from a control rat (HE, ×200); B: Lung from a rat of pulmonary fibrosis filled with inflammatory cell infiltration, thickened alveolar walls, damaged alveolar structures, increased fibronodules and macrophage aggregation (HE, ×200). C: Lung from a control rat (Masson, ×200); D: Lung from a rat of pulmonary fibrosis filled with more and thicker collagen fibers, damaged alveolar septum accompanied by a small amount of inflammatory cell infiltration, forming a diffuse fibrosis nodular changes (Masson, ×200).

collecting duct acid secretion (5 proteins), complement and coagulation cascades (5 proteins) and glutathione metabolism (4 proteins). Additional, ECM-receptor interactions (4 proteins) and antigen processing and presentation (4 proteins) were also observed to be significantly enriched. It was observed that the DEPs were involved in physical or functional interactions to constitute a network through STRING database analysis (Fig 6). The PPI network analysis found that some DEPs interact with each other, such as CD14-CD68-Atp6v0c-Paps-Gm2a-Gns, Cldn5-Esam-CD34-Col1a1-Itga8-Col4a3 and Rac2-Cyba-Ncf2-Ncf4. These key focus hubs have important biological functions in biological regulation, oxidative stress, enzyme activity, cell migration and motility, lysosome, biological adhesion, expose to stimulus, receptor binding, etc.

## Discussion

TMT technique is used for quantitative proteomics with high throughput and high reproducibility. PRM analysis was used for validating the accuracy and reproducibility of the proteomic

**Table 1. Effects of silica on lung alveolitis and pulmonary fibrosis (means ± SD).**

| Group | Number | Alveolitis score | Pulmonary fibrosis score |
|---|---|---|---|
| Control group | 10 | 0 | 0 |
| Model group | 10 | 1.30±0.48 [a] | 1.80±0.42[a] |

[a]P<0.05, compared with the control group.

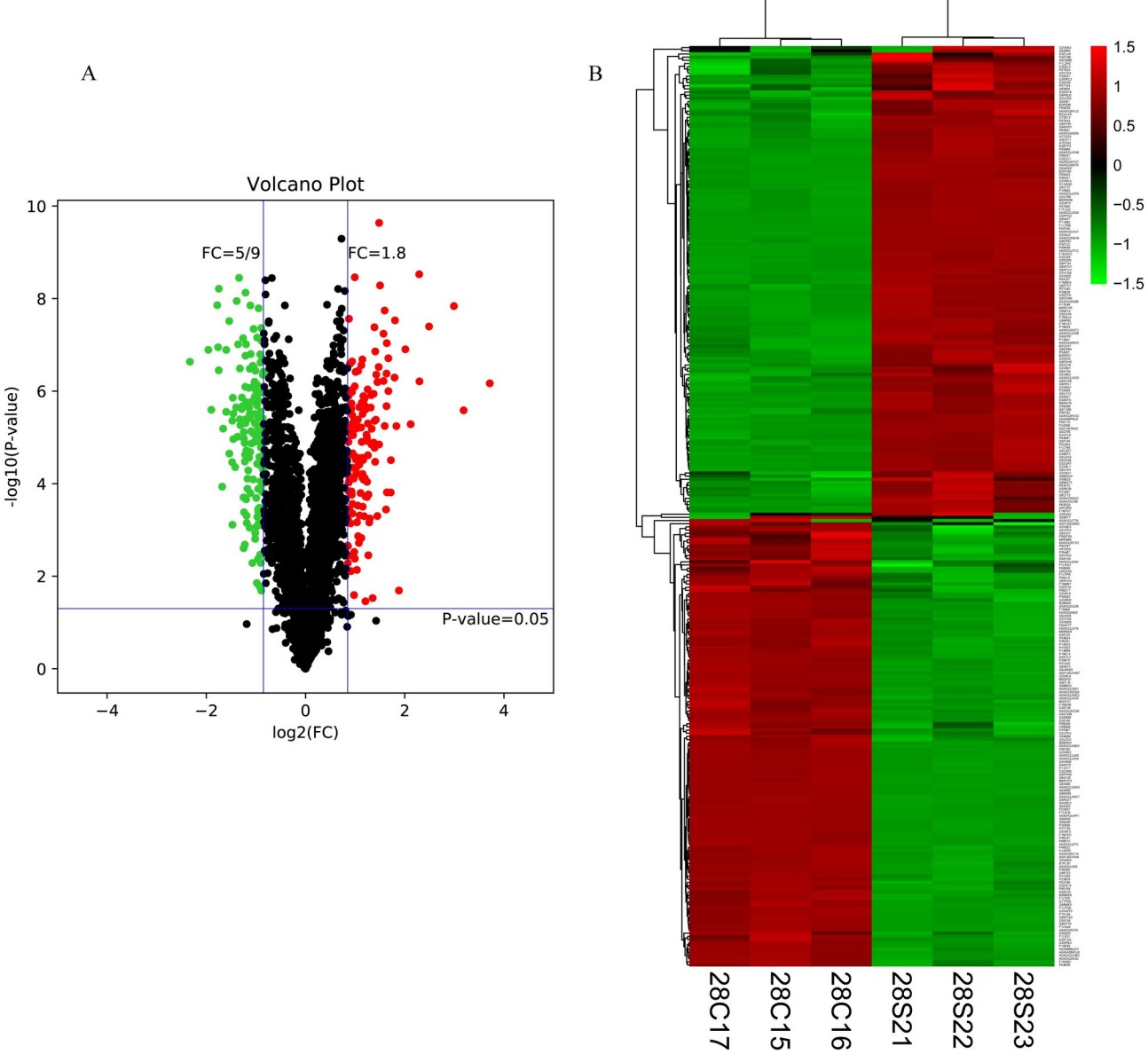

**Fig 2. DEPs between the control and model groups.** A shows volcano plot of proteins. The threshold set for DEPs was a fold change (FC) ≥1.8 and p value < 0.05. 145 proteins are up-regulated (red) and 140 proteins are down-regulated (green). B shows heat map of DEPs between the control and model groups, with folds > ± 1.8 and p value < 0.05. Each column represents a sample and each row represents a significant protein. [28S21], [28S22] and [28S23] represent model samples; [28C15], [28C16] and [28C17] represent control samples. 285 proteins were found to be significantly differentially expressed.

data. Our results showed that pulmonary fibrosis was induced by a single exposure to silica particles by intratracheally instillation in rats. Two hundred and eighty-five DEPs were identified between the control and silicosis model groups (Fig 1). We selected six proteins for verification using PRM and the results showed a similar expression trend with TMT (Fig 3), suggesting the reliability of our TMT analysis. The DEPs mainly enriched in the pathways of phagosome, lysosome, oxidative stress and ECM-receptor interactions.

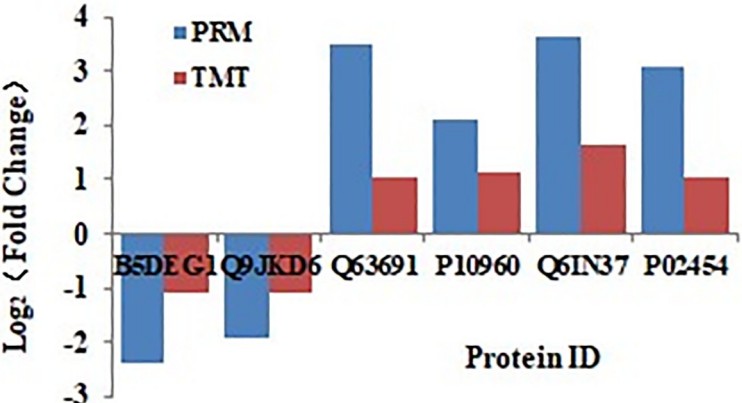

**Fig 3. PRM verification of proteins identified by TMT analysis.** Six proteins were selected for validation of the TMT data. The abscissa represents the protein ID. The ordinate represents the log2 (Fold change) of the DEPs measured by TMT and PRM. The trends of the level of expression of these proteins obtained by PRM were similar to TMT.

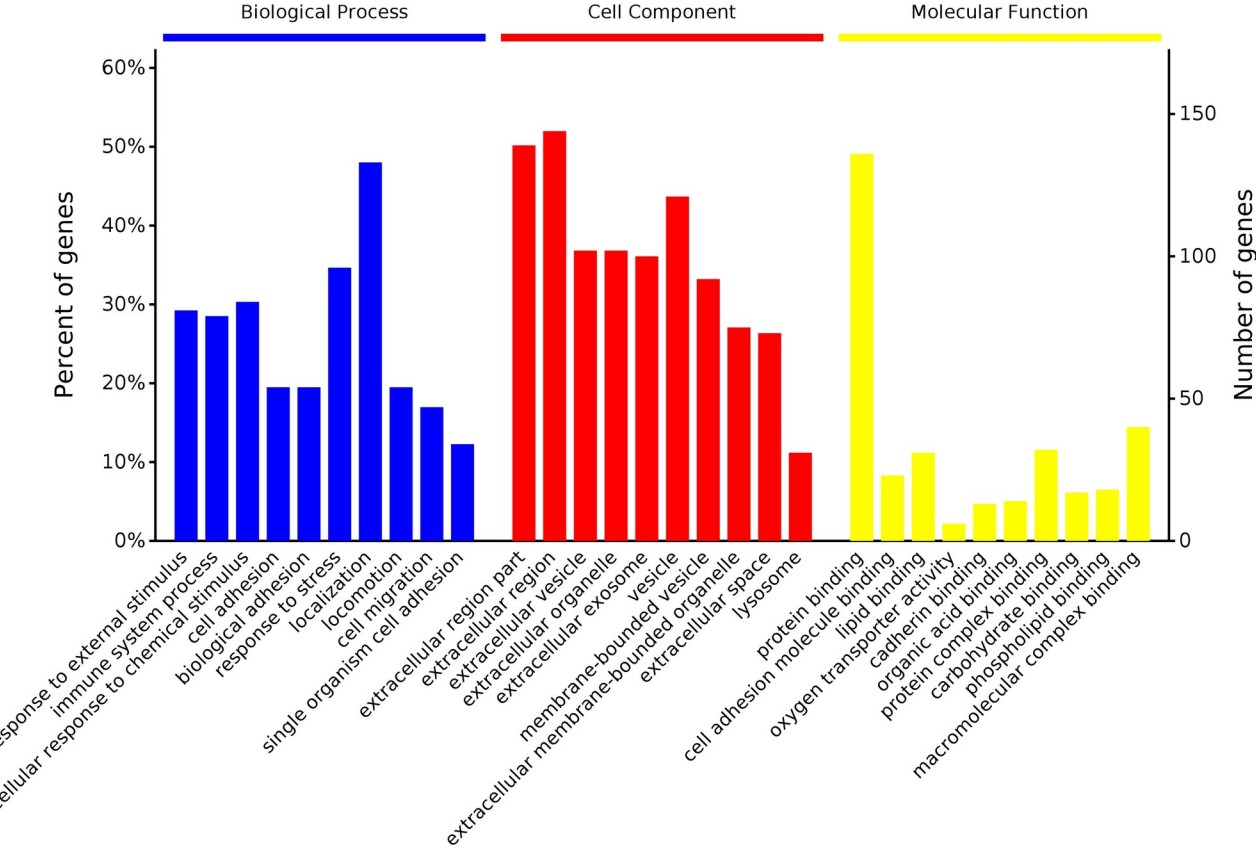

**Fig 4. GO analysis of 285 DEPs between the control and model groups.** The top ten biological process categories, cellular component categories and molecular function are presented.

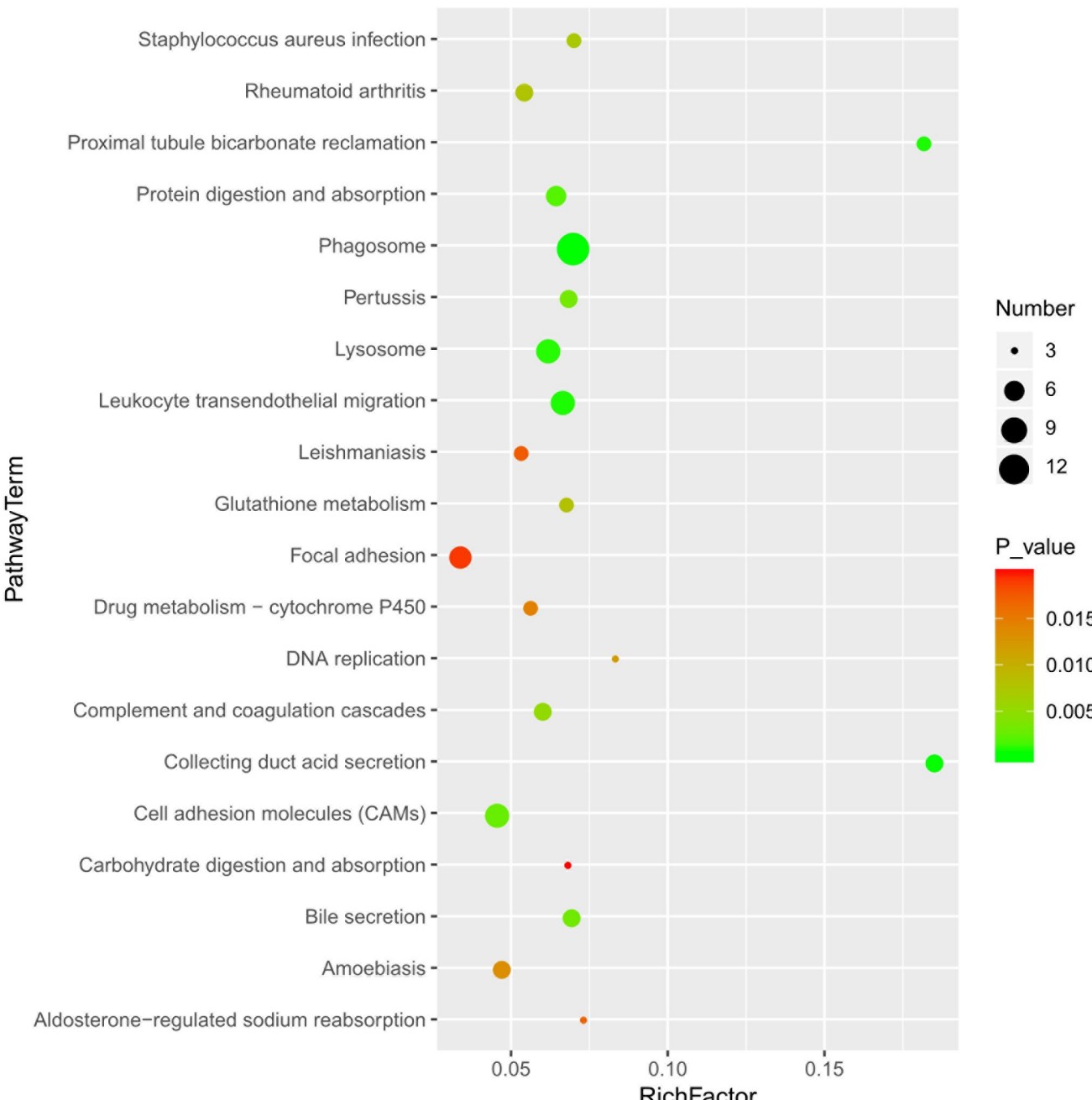

**Fig 5. Scatter diagram of the enriched KEGG pathways of the 285 DEP.** The top twenty terms are shown. Degree of enrichment was measured by Rich factor, Q value, and the number of genes enriched in one pathway. The Rich factor is the ratio of the number of differentially expressed genes enriched in one pathway and the total annotation number. The greater the Rich factor value, the higher the degree of enrichment. The Q value is a variant of a p value, for which lower numbers equate to significant enrichment. The Y-axis represents the name of the pathway and the X-axis represents the Rich factor. The point size is the number of differentially expressed genes in one pathway, and the color of the point indicates the range of the Q value.

## Silica exposure influences phagosome

Inhaled silica dust is predominantly phagocytosed by alveolar macrophage (AM) when it enters the pulmonary alveolus [25]. In this study, phagosome proteins such as SFTPA1, RT1-Ba, CD14, SCARB1 and SEC61B were upregulated in the lung of silica-exposed rats.

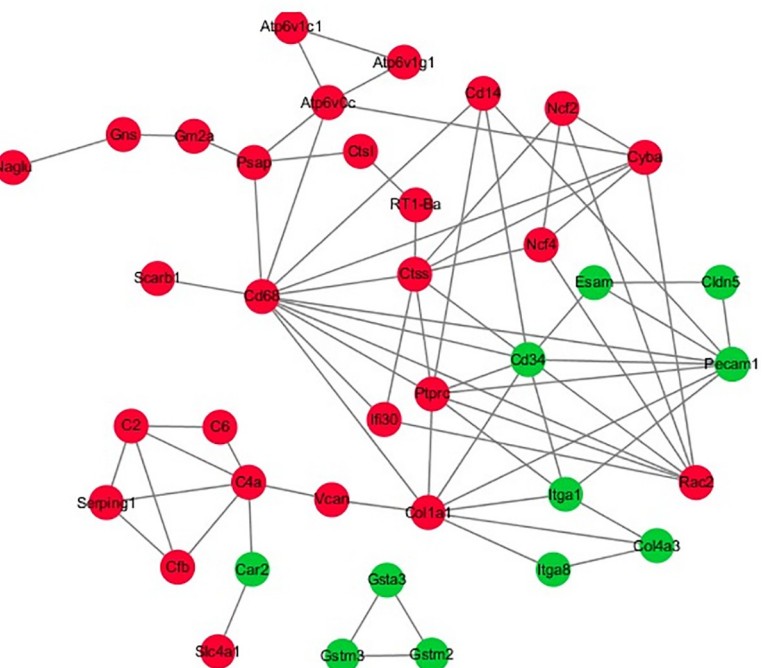

**Fig 6. The interacted network of proteins was analyzed by STRING software.** Adual-color code was used, with red and green indicating up- and down regulation, respectively.

Macrophages are innate immune cells with various types of receptors such as Fc receptors, scavenger receptors, and Toll-like receptors (TLRs), of which TLRs are crucial to macrophage phagocytosis. Silica particle, recognized as pathogen-associated molecular patterns by the innate immune system, binds the membrane-bound TLRs to active two different signaling cascades: the "myeloid differentiation primary response gene 88 (MyD88)-dependent" and the "Toll/ interleukin 1 receptor domain-containing adaptor-inducing interferon-β (TRIF)-dependent" cascades. These signaling cascades induce the activation of nuclear factor κB (NF-κB) and interferon regulatory factor 3 (IRF3). Finally, inflammatory cytokines such as tumor necrosis factor (TNF)-α and interleukin (IL)-1β are produced to promote fibrosis [26]. CD14 has been shown to be required for TLR4 endocytosis to activate downstream signaling [27]. The core fucosylation deficiency in CD14 suppressed the endocytosis of TLR4 and impaired TLR4 signaling in mouse embryonic fibroblasts [28]. Alveolar macrophage may phagocytose silica particles through TLR4-mediated recognition. Additionally, Scavenger Receptor Class B Member 1 (SCARB1) is a silica receptor associated with canonical inflammasome activation [29]. Phagocytosed silica particles cannot be digested in phago-lysosomes, which induces lysosomal stress and activates NLRP3 inflammasomes, followed by progressive lung fibrosis [30].

## Silica exposure influences lysosome

Inhaled silica dust disrupted lysosomes, which released lysosomal cathepsins (Cats) and other hydrolases into the cytosol [31]. In this study, we confirmed Cat D, S, H, L and PSAP were significantly upregulated in the lung of silica-exposed rats. Upregulated Cat S, L, B and K were also observed in the lung of silica-exposed mice [32]. Cat B contributes to lung fibroblast differentiation into myofibroblasts by triggering TGF-β1-driven canonical SMAD pathway. Inhibition of Cat B diminished α-SMA expression and delayed lung fibroblast differentiation [33],

and also reduced hepatic inflammation, collagen deposition and fibrogenesis [34]. Conversely, Cat K could inactivate TGF-β1 and restrict excessive ECM deposition to control lung fibrosis [35], and Cat S may proteolytically inactivate Cat K and thus would control its collagenolytic or elastinolytic activity [36]. Additionally, multiple Cathepsins such as Cat B, L, C, S and X promote pro-IL-1β synthesis and NLRP3-mediated IL-1β activation in murine macrophages [37]. Prosaposin (PSAP) is a precursor for four sphingolipid activator proteins known as saposins A-D, which serve as activators for lysosomal hydrolases [30]. PSAP can reverse the inhibitory effects of Cystatin C (CST3) on Cathepsins by forming a complex that changes the conformational properties [38]. In prostate cancer cell, downregulation of PSAP decreased b1A-integrin expression, its cell-surface clustering, and adhesion to basement membrane proteins. Cat D expression and proteolytic activity, migration, and invasion were also significantly decreased in *PSAP* knock-down cells [39]. Downregulation of PSAP might be contribute to silicosis therapy.

## Silica exposure influences oxidative stress

Silicosis is a disease associated with oxidative stress. In AM, ROS was mainly generated by NADPH oxidase (NOX) from alveolar macrophages. In this study, we conformed the subunits of NADPH oxidase complex such as $NOX_2$ (gp91$^{phox}$), p22$^{phox}$, p47$^{phox}$, p40$^{phox}$ and p67$^{phox}$ were upregulated in the lung of silica-exposed rats. By activating p38 MAPK signaling pathway ROS disrupted lung endothelial integrity and increased vascular hyperpermeability [40], which created a pro-fibrotic intra-alveolar environment to promote several profibrotic responses, such as intra-alveolar coagulation and provisional matrix establishment [41]. CLDN5, a marker for endothelium tight junctions and permeability [42], predominantly expressed in the cell-cell junctions of alveolar endothelial cells and played critical roles in the pulmonary endothelial barrier. Downregulation of CLDN5 was associated with disrupted endothelium tight junctions in bleomycin-induced pulmonary fibrosis, and which may be involved in epithelial-mesenchymal transition (EMT). TGF-β also disrupted the alveolar epithelial and endothelial tight junctions by downregulating CLDN5 expression [43]. In cardiac fibroblasts and endothelial cells TGF-β also induced COL1A1 expression by downregulating CLDN5 expression, which also promoted macrophage infiltration and pro-fibrotic responses [44]. We Confirmed CLDN5 was downregulated in the lung of silica-exposed rats in this study. Endothelial hyperpermeability induced by oxidative damage may contribute to silica-induced pulmonary fibrosis. Therefore, a therapeutic approach of limiting the extent of vascular leak may be an effective strategy for treating silicosis.

## Silica exposure influences Extracellular Matrix (ECM)

The primary pathological characteristic of silicosis is the imbalance of extracellular matrix anabolism and catabolism. MMPs degrade all ECM components as well as divers nonmatrix proteins including cytokines, chemokines, and receptors, but the catalytic activity of MMPs can be compromised by the tissue inhibitor of metalloproteinases (TIMP) family. In this study, upregulated MMP-8 and downregulated TIMP-3 were identified in the lung of silica-exposed rats. MMP-8 can degrade basement membrane and extracellular proteins, causing airway disruption and remodeling in chronic obstructive pulmonary disease (COPD) [45]. TIMP-3 has been recognized as a key regulator in lung homeostasis, which plays a versatile part in the development of inflammation as well as fibrosis, rather than merely acting through the restriction of ECM degradation. More severe fibrosis occurs in bleomycin-injured TIMP3-deficient mice [46].

Integrin α8 (ITGA8), an important component of ECM-receptor interaction pathway, played important roles in the expression of extracellular matrix components [47]. ITGA8 may participate in the degradation of extracellular matrix, including collagen type XI alpha 1, aggrecan, collagen type VI alpha 1 [48]. Additionally, ITGA8 attenuated tubulointerstitial fibrosis by regulation of TGF-β /Smad2/3 signaling, fibroblast activation and immune cell infiltration [49]. Deficiency of ITGA8 worsened tubulointersititial fibrosis [50] and delayed healing in a model of glomerulonephritis [51]. In the lung, ITGA8 expression was restricted to interstitial stromal cells, and that was increased in bleomycin-induced fibrosis. ITGA8 deletion increased COL1A1 production during lung fibrosis in vitro, but did not affect pulmonary fibrosis in the bleomycin animal model [52]. We confirmed ITGA8 was decreased in silica-induced pulmonary fibrosis, and the role of ITGA8 in silica-induced needs further research.

## Conclusion

In summary, we found some proteins which are closely relevant to the occurrence and development of silicosis using TMT coupled with PRM technology. Most proteins were enriched in immune system processes, oxygen transporter activity, phagosome, lysosome and ECM-receptor interactions. These findings will further provide useful clues to elucidate pathogenesis of silicosis and reveal more potential therapeutic targets.

## Supporting information

**S1 Table. All peptide sequences identified through TMT-based quantitative proteomics.**
(XLSX)

**S2 Table. 3099 proteins identified and quantified through TMT-based quantitative proteomics.**
(XLSX)

**S3 Table. 145 upregulated proteins in lungs of silica-exposed rats.**
(XLSX)

**S4 Table. 140 downregulated proteins in lungs of silica-exposed rats.**
(XLSX)

## Acknowledgments

The authors are grateful to Prof. Martin F Lavin for assistance with the manuscript.

## Author Contributions

**Conceptualization:** Cunxiang Bo, Qiang Jia.

**Data curation:** Cunxiang Bo, Zhenling Zhang.

**Formal analysis:** Cunxiang Bo, Kai Liu.

**Investigation:** Juan Zhang, Kai Liu, Zhongjun Du.

**Methodology:** Linlin Sai, Yu Zhang, Gongchang Yu, Cheng Peng.

**Project administration:** Cunxiang Bo, Qiang Jia.

**Resources:** Xiao Geng, Juan Zhang.

**Supervision:** Qiang Jia, Hua Shao.

**Validation:** Linlin Sai, Gongchang Yu.

**Visualization:** Cunxiang Bo, Xiao Geng, Yu Zhang.

**Writing – original draft:** Cunxiang Bo, Xiao Geng.

**Writing – review & editing:** Cheng Peng, Qiang Jia.

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
