## [Decision Letter · Decision Letter 0]

23 Jun 2020

PONE-D-20-07393

Comparative proteomic analysis of silica-induced pulmonary fibrosis in rats based on tandem mass tag (TMT) quantitation technology

PLOS ONE

Dear Dr. Bo,

Thank you for submitting your manuscript to PLOS ONE. After careful consideration, we feel that it has merit but does not fully meet PLOS ONE’s publication criteria as it currently stands. Therefore, we invite you to submit a revised version of the manuscript that addresses the points raised during the review process.

Both of the reviewers believed the study was interesting and a valuable addition to the literature.  However there were a number of issues that need to be address as listed by the reviewers.  Additionally,  there were a number of grammatical issues that need to be addressed.

We look forward to receiving your revised manuscript.

Kind regards,

Yuqin Yao

Academic Editor

PLOS ONE

Journal Requirements:

2. Please send accession numbers for mass spec data.

3. We noticed minor instances of text overlap with the following previous publication(s), which need to be addressed:

https://www.sciencedirect.com/science/article/abs/pii/S0378427419300852?via%3Dihub

https://www.amjmedsci.org/article/S0002-9629(16)30631-0/fulltext

4. To comply with PLOS ONE submissions requirements, please provide methods of sacrifice in the Methods section of your manuscript.'

In your revision please ensure you cite all your sources (including your own works), and quote or rephrase any duplicated text outside the methods section. Further consideration is dependent on these concerns being addressed.

We thank OeBiotech Corporation (Shanghai, China) for supporting the high throughpu.

This study was supported by the Innovation Project of Shandong Academy of Medical

Sciences, Academic promotion programme of Shandong First Medical University

(2019QL001), the Department of Science and Technology of Shandong Province

(2018GSF118212, 2018GSF121007), Natural Science Foundation of Shandong

(ZR2017YL001), China Coal Miner Pneumoconiosis Prevention Treatment Foundation

(201915J039), the National Nature Science Foundation of Chian (No.81872603,

81600293). The funders had no role in study design, data collection and analysis,

decision to publish, or preparation of the manuscript.

Additionally, because some of your funding information pertains to [commercial funding//patents], we ask you to provide an updated Competing Interests statement, declaring all sources of commercial funding.

In your Competing Interests statement, please confirm that your commercial funding does not alter your adherence to PLOS ONE Editorial policies and criteria by including the following statement: "This does not alter our adherence to PLOS ONE policies on sharing data and materials.” as detailed online in our guide for authors  http://journals.plos.org/plosone/s/competing-interests.  If this statement is not true and your adherence to PLOS policies on sharing data and materials is altered, please explain how.

Please include the updated Competing Interests Statement and Funding Statement in your cover letter. We will change the online submission form on your behalf.

Reviewers' comments:

Reviewer's Responses to Questions

**Comments to the Author**

1. Is the manuscript technically sound, and do the data support the conclusions?

Reviewer #1: Partly

Reviewer #2: No

2. Has the statistical analysis been performed appropriately and rigorously? 

Reviewer #1: No

Reviewer #2: I Don't Know

3. Have the authors made all data underlying the findings in their manuscript fully available?

Reviewer #1: Yes

Reviewer #2: No

4. Is the manuscript presented in an intelligible fashion and written in standard English?

Reviewer #1: Yes

Reviewer #2: No

5. Review Comments to the Author

Reviewer #1: Bo et al describe a TMT-based LC-MS/MS discovery experiment to better understand the molecular underpinnings of their mouse silicosis model. After applying rudimentary p-value and fold-change cutoffs, the authors describe 285 differentially expressed proteins, 6 of which they went on to confirm via targeted PRM analysis. They subsequently plugged these differentially expressed proteins into a variety of online tools including, DAVID, GO, KEGG, and STRING.

While the differential proteins that they identify and the contextualization of those into what is known about the etiology of the disease is reasonable, the overarching weakness of this paper is that the only gating criteria for differential expression was a fold-change cut-off of 1.8 and a p-value of 0.05. I can’t find anywhere that they attempted to apply a multiple testing correction to generate an estimate of what the false discovery rate was in their 285 proteins. With nearly 3100 identified proteins, it is likely that a large proportion of those are false positives. If so, then all of the downstream analyses using the online tools may be compromised by that high false positive rate.

It seems like the field would benefit from the publication of solid proteomics data on this model system of silicosis. Even without dose response or time-course data, researchers in the field would likely find the up- and down-regulation data useful. However, to publish without a more rigorous workup the data, potentially limits its utility. There might even be different thresholds utilized because different downstream tools might have more tolerance for false positives (e.g. DAVID).

Minor criticisms:

Figure 2 – the heat-map is so small as to be difficult to see. One potentially interesting aspect is that a subset of the clades of proteins do not behave consistently across the replicates. Are these false positives or do they represent a subset of proteins that are showing differential disease response in those samples that is potentially relevant to the scope of the phenotype being observed.

Figure 3 – error bars for the measurements are needed.

Figure 5 – the scaling size of the circles is too small to visualize easily. If each were approximately 5 to 10-fold larger, the figure would be more readily interpretable.

Reviewer #2: Cunxiang Bo et al. reported that silica-induced pulmonary fibrosis specimens obtained from the rat model were subjected to proteome analysis using TMT method, and Gene Ontology and KEGG pathway analysis were performed. Furthermore, the results obtained by TMT method were confirmed by PRM method.

The importance to determine the etiology and to develop early diagnosis for pulmonary fibrosis is clearly stated in the introduction (line 51-53). They also say that they investigate proteins that could lead to diagnosis, prevention, and treatment (lines 75-77). However, there are too many candidates (lines 25-32) and it is difficult to use their results as a reference because we do not know whether we should focus some of them as a reference. They just listed the broad results obtained in this study. My suggestions are as follows: They can have total 20 rats in both groups at least as their capability, 3 time points and 3 rats per point as a time course study would be conducted. This would allow them to narrow down the candidate proteins. Alternatively, the drug dosage could be changed for comparison. Although only 6 rats out of 20 were measured in this study (because the TMT reagent is 6-plex?), they should put those 20 rats to good use (time course or different dosage).

According to the authors, 285 proteins were differentially expressed in TMT method, of which only 6 proteins in PRM method were reproduced. If this understanding is correct, there is no need to analyze the data obtained by TMT method, because other 279 proteins were not reproducible. If authors want to argue that TMT reproducibility is poor and TMT results should always be confirmed by a different method, then it should be stated in the text.

They stated that a total of 3099 proteins were identified, but the protein names, identified peptide sequences, retention time, m/z, identification score and quantitative values should be provided in a supplemental information. In addition, it should be listed whether it was identified from TMT method or PRM method.

The official name of the abbreviation such as DTT, TEAB and IAA should also be included.

To ensure the reliability of the raw data in this study, the LC/MS data should be uploaded to iProX database (https://www. iprox.org/)(Beijing Proteome Research Center, Beijing, China).

6. PLOS authors have the option to publish the peer review history of their article (what does this mean?). If published, this will include your full peer review and any attached files.

Reviewer #1: No

Reviewer #2: No

---

## [Author Response · Author response to Decision Letter 0]

30 Jul 2020

Dear editors,

Thank you for your and reviewers’ valuable comments. We have updated the manuscript accordingly by addressing all points from you and the reviewers and corrected linguistic errors when necessary. Our revisions and response to the comments are listed as follows. All the changes are marked in “Revised Manuscript with Track Changes”. 

Journal Requirements:

Responses: We've adjusted the manuscript according to the style requirements of PLOS ONE including those for file naming. The legends of Fig 2 ang Fig 3 were rephrased (line 226-230 and line 245-248). 

2. Please send accession numbers for mass spec data.

Responses: The mass spectrometry proteomics data have been deposited to the ProteomeXchange Consortium (http://proteomecentral.proteomexchange.org) via the iProX partner repository with the dataset identifier PXD020625（line 187-190）.

3. We noticed minor instances of text overlap with the following previous publication(s), which need to be addressed:

https://www.sciencedirect.com/science/article/abs/pii/S0378427419300852?via%3Dihub. https://www.amjmedsci.org/article/S0002-9629(16)30631-0/fulltext

Responses: All the overlapping text is mainly the methods section. We used the same methods to establish silicosis rat model and assess the pulmonary fibrosis. The overlaping text in introduction (line 61-64), method (line 90-97) and results (line 202-208) have been revised and the two papers were also cited in our revised manuscript.

4. To comply with PLOS ONE submissions requirements, please provide methods of sacrifice in the Methods section of your manuscript.

In your revision please ensure you cite all your sources (including your own works), and quote or rephrase any duplicated text outside the methods section. Further consideration is dependent on these concerns being addressed.

Responses: All rats were sacrificed by CO2 anesthesia in our study (line 86-87) which is in accordance with animal welfare and ethics regulated by our institute. We have revised the wording and cited all sources. The order of references in the manuscript was modified correspondingly.

5. PLOS requires an ORCID iD for the corresponding author in Editorial Manager on papers submitted after December 6th, 2016. Please ensure that you have an ORCID iD and that it is validated in Editorial Manager. 

Responses：The ORCID iD of the corresponding author has been validated in Editorial Manager.

6. Thank you for stating the following in the Acknowledgments Section of your manuscript: We thank OeBiotech Corporation (Shanghai, China) for supporting the high throughpu. 

We note that you have provided funding information that is not currently declared in your Funding Statement. However, funding information should not appear in the Acknowledgments section or other areas of your manuscript. We will only publish funding information present in the Funding Statement section of the online submission form. Please remove any funding-related text from the manuscript and let us know how you would like to update your Funding Statement. 

Additionally, because some of your funding information pertains to [commercial funding//patents], we ask you to provide an updated Competing Interests statement, declaring all sources of commercial funding.

In your Competing Interests statement, please confirm that your commercial funding does not alter your adherence to PLOS ONE Editorial policies and criteria by including the following statement: "This does not alter our adherence to PLOS ONE policies on sharing data and materials.” as detailed online in our guide for authors http://journals.plos.org/plosone/s/competing-interests. If this statement is not true and your adherence to PLOS policies on sharing data and materials is altered, please explain how.

Responses：The OeBiotech Corporation offers only high-throughput technology platforms without any funding support. To avoid any misunderstanding, the acknowledgement about the OeBiotech Corporation was removed from the manuscript. Funding information is listed as follows.

Funding Statement：This study was supported by Natural Science Foundation of Shandong (ZR2017YL001), the Innovation Project of Shandong Academy of Medical

Sciences, Academic promotion programme of Shandong First Medical University

(2019QL001), the Department of Science and Technology of Shandong Province

(2018GSF118212, 2018GSF121007), China Coal Miner Pneumoconiosis Prevention Treatment Foundation (201915J039), the National Nature Science Foundation of Chian (No.81872603, 81600293). The funders had no role in study design, data collection and analysis, decision to publish, or preparation of the manuscript.

Competing Interests Statement: Our research is supported only by government funding. The authors have read the journal’s policy and declare that they have no known competing financial interests or personal relationships that could have appeared to influence the work reported in this paper. 

Competing Interests Statement and Funding Statement were also stated in our cover letter (line 24-36).

Reviewers' comments:

Reviewer #1:

Comments: While the differential proteins that they identify and the contextualization of those into what is known about the etiology of the disease is reasonable, the overarching weakness of this paper is that the only gating criteria for differential expression was a fold-change cut-off of 1.8 and a p-value of 0.05. I can’t find anywhere that they attempted to apply a multiple testing correction to generate an estimate of what the false discovery rate was in their 285 proteins. With nearly 3100 identified proteins, it is likely that a large proportion of those are false positives. If so, then all of the downstream analyses using the online tools may be compromised by that high false positive rate.

It seems like the field would benefit from the publication of solid proteomics data on this model system of silicosis. Even without dose response or time-course data, researchers in the field would likely find the up- and down-regulation data useful. However, to publish without a more rigorous workup the data, potentially limits its utility. There might even be different thresholds utilized because different downstream tools might have more tolerance for false positives (e.g. DAVID).

Responses: TMT is one of the most sensitive techniques used in comparative proteomics with high throughput and high reproducibility. To reduce the probability of false peptide identification, we set various parameters to ensure the accuracy of proteins identification, including a peptide mass tolerance of ± 10 ppm, variable modifcations of oxidation (M), a fragment mass tolerance of 0.02 Da，decoy as the database pattern and a peptide false discovery rate (FDR) of ≤0.01. FDR is the metric for global confidence assessment of a large-scale proteomics dataset [Suruchi Aggarwal, 2016, doi: 10.1007/978-1-4939-3106-4_7.]. For protein quantization, the protein was required to contain at least one unique peptide. The quantitative protein ratios were weighted and normalized by the median ratio in Mascot (references added) [Li L, et al., 2019, doi: 10.2147/JPR.S185916.eCollection2019, and Wu X, et al., 2019 doi: 10.1128/AAC.00160-19]. The ratio of mean expression between model and control was based on three biological replicates. The significant difference in the levels of proteins expression between model and control was determined by independent sample t-test. Proteins with P≤0.05 and FC > ±1.8 were considered as DEPs. The above including references has been added to our manuscript (line 157-164 and line 195-199). All these efforts minimized the false positive rate.

Comments: Figure 2 – the heat-map is so small as to be difficult to see. One potentially interesting aspect is that a subset of the clades of proteins do not behave consistently across the replicates. Are these false positives or do they represent a subset of proteins that are showing differential disease response in those samples that is potentially relevant to the scope of the phenotype being observed.

Responses: The small heat-map is because of a large number of differential proteins. The figure can be enlarged without losing the resolution. It is possible that a few individual proteins have inconsistent expression trends within the biological replicates because the samples are not from the same rat. The similar contents can also be found in the following publications:

http://attach.pubtsg.com:8088/attach/show?query=BsKDYnAxLMohFSVwjwx7e6gXeSPlK0noRWf_cl7Jz55cAxt4KyPE0-1G8dE&view=true&type=.pdf

http://attach.pubtsg.com:8088/attach/show?query=c4Ss5EcCEUJdZ4HEOailZEGRa6_fk80ol2_oURTBpDdeFedxCwb6yHM9nOc&view=true&type=.pdf

Comments: Figure 3 – error bars for the measurements are needed.

Responses: In figure 3 the abscissa represents the protein ID, the ordinate represents the log2 (Fold change) of the DEPS measured by iTRAQ and PRM. Error bars are not applicable because fold change is the ratio of mean protein expression between model and control group.

Comments: Figure 5 – the scaling size of the circles is too small to visualize easily. If each were approximately 5 to 10-fold larger, the figure would be more readily interpretable.

Responses: The scaling size of the circles has been magnified in our revised manuscript. 

Reviewer #2: 

Comments: The importance to determine the etiology and to develop early diagnosis for pulmonary fibrosis is clearly stated in the introduction (line 51-53). They also say that they investigate proteins that could lead to diagnosis, prevention, and treatment (lines 75-77). However, there are too many candidates (lines 25-32) and it is difficult to use their results as a reference because we do not know whether we should focus some of them as a reference. They just listed the broad results obtained in this study. My suggestions are as follows: They can have total 20 rats in both groups at least as their capability, 3 time points and 3 rats per point as a time course study would be conducted. This would allow them to narrow down the candidate proteins. Alternatively, the drug dosage could be changed for comparison. Although only 6 rats out of 20 were measured in this study (because the TMT reagent is 6-plex?), they should put those 20 rats to good use (time course or different dosage).

Responses: We used a single silica exposure by intratracheally instillation with 50 mg/mL silicon dioxide (1 mL per rat) to establish silicosis rat model successful, which is a commonly used method and has been used in our previous articles [Sai L, et al., 2019, doi: 10.1016/j.toxlet.2019.04.003., Guo, J, et al., 2019, doi: 10.1016/j.toxlet. 2018.10.019.]. The study is focused on the silicosis but not the toxic effects of silica particles. Therefore, it's not necessary to use different doses to build animal models. In our study, obvious pulmonary fibrosis was observed on the 28th day after silica exposure, which were assessed by histopathologic examination. We also found that all rats in the model group showed significant pulmonary fibrosis with good repeatability. Three lungs from each group were randomly selected for quantitative proteomic analysis, which achieved the minimum requirements for TMT and biological replicates. Based on the reviewer’s comments, we rephrased the abstract (line 23-31).

Thanks for reviewer's opinion, we will perform the quantitative proteomics analysis of rat lung tissues at different time points to identify dynamic proteins in silicosis in the future. 

Comments: According to the authors, 285 proteins were differentially expressed in TMT method, of which only 6 proteins in PRM method were reproduced. If this understanding is correct, there is no need to analyze the data obtained by TMT method, because other 279 proteins were not reproducible. If authors want to argue that TMT reproducibility is poor and TMT results should always be confirmed by a different method, then it should be stated in the text. 

Responses: TMT technique is used for quantitative proteomics because of its high throughput and reproducibility. In our study 285 proteins were differentially expressed in TMT method, which with P≤0.05 and fold changes> ±1.8. We used PRM analysis to validate the accuracy and reproducibility of the proteomic date, for which we select six proteins with larger FC ＞2.0 value and potentially important biological functions related to inflammation or fibrosis for PRM identification. These proteins showed exactly the same trend of expression as those observed in TMT. The high consistency between the results of PRM and iTRAQ indicated the accuracy and reproducibility of our proteomic data.

Comments: They stated that a total of 3099 proteins were identified, but the protein names, identified peptide sequences, retention time, m/z, identification score and quantitative values should be provided in a supplemental information. In addition, it should be listed whether it was identified from TMT method or PRM method.

Responses: 3099 identified proteins from TMT were provided in a supplemental information（Table S1-2）in our revised manuscript (line 586-591).

Comments: The official name of the abbreviation such as DTT, TEAB and IAA should also be included.

Responses: We added the official names to all the abbreviations in our revised manuscript (line 71, 72,111,122-124).

Comments: To ensure the reliability of the raw data in this study, the LC/MS data should be uploaded to iProX database (https://www. iprox.org/) (Beijing Proteome Research Center, Beijing, China).

Responses: The mass spectrometry proteomics data have been deposited to the ProteomeXchange Consortium (http://proteomecentral.proteomexchange.org) via the iProX partner repository with the dataset identifier PXD020625.

We thank the reviewers for their constructive advices and helpful comments that definitively helped to improve our manuscript. We believe the carefully revised manuscript is much improved and suitable for publication.

---

## [Decision Letter · Decision Letter 1]

27 Aug 2020

PONE-D-20-07393R1

Comparative proteomic analysis of silica-induced pulmonary fibrosis in rats based on tandem mass tag (TMT) quantitation technology

PLOS ONE

Dear Dr. Bo,

Thank you for submitting your manuscript to PLOS ONE. After careful consideration, we feel that it has merit but does not fully meet PLOS ONE’s publication criteria as it currently stands. Therefore, we invite you to submit a revised version of the manuscript that addresses the points raised during the review process.

ACADEMIC EDITOR:

1. Please pay attention to the journal name in line 7 of cover letter. Please ensure this manuscript has not been published elsewhere and is not under consideration by another journal.

2. The results of TMT proteomics analysis can provide a new idea for the study of the pathogenesis of silicosis. However, the results of the manuscript only show the primary results of TMT analysis, and do not carry out in-depth data analysis, which is the biggest shortcoming of this manuscript. It is suggested that the authors make full use of the data to further explain the pathogenesis of silicosis. Considering that there are few reports on the proteomics of silicosis, the publication of this manuscript in Plos One is still of some significance.

We look forward to receiving your revised manuscript.

Kind regards,

Yuqin Yao

Academic Editor

PLOS ONE

Reviewers' comments:

Reviewer's Responses to Questions

**Comments to the Author**

1. If the authors have adequately addressed your comments raised in a previous round of review and you feel that this manuscript is now acceptable for publication, you may indicate that here to bypass the “Comments to the Author” section, enter your conflict of interest statement in the “Confidential to Editor” section, and submit your "Accept" recommendation.

Reviewer #1: (No Response)

Reviewer #2: All comments have been addressed

2. Is the manuscript technically sound, and do the data support the conclusions?

Reviewer #1: Partly

Reviewer #2: Yes

3. Has the statistical analysis been performed appropriately and rigorously? 

Reviewer #1: No

Reviewer #2: Yes

4. Have the authors made all data underlying the findings in their manuscript fully available?

Reviewer #1: Yes

Reviewer #2: Yes

5. Is the manuscript presented in an intelligible fashion and written in standard English?

Reviewer #1: Yes

Reviewer #2: Yes

6. Review Comments to the Author

Reviewer #1: While the textual changes and figure modifications have improved the overall quality of the manuscript. The authors have fundamentally failed to address my primary concern in their initial submission – namely false discoveries in their list of differentially expressed proteins (DEPs). In their response to reviewer comments, they highlight why they are confident in their peptide spectrum matches and the protein inferences are of high confidence. Those were not the issue. The question is how many of the proteins that pass the threshold of a simple t-test p<0.05 and a FC >1.8 are false positive DEPs. With >3000 tests, you would expect a large proportion of the proteins that passed a p<0.05 threshold to be random hits (~150). While there is some debate about how best to utilize multiple testing corrections (e.g. Benjamini and Hochberg) in small magnitude effect measurements like iTRAQ/TMT (https://onlinelibrary.wiley.com/doi/pdf/10.1002/pmic.201600044), it is something that should not just be ignored. If BH ends up being too aggressive, there are other options for at least characterizing what the likely FDR would be using a process like in DAPAR/Prostar (https://onlinelibrary.wiley.com/doi/pdf/10.1002/pmic.201600044) or additional filtering using a z-score (https://www.sciencedirect.com/science/article/pii/S187439191530186X?via%3Dihub). The authors could even run their data through existing analysis pipelines like MSstatsTMT (https://www.mcponline.org/content/early/2020/07/17/mcp.RA120.002105) or the PAW pipeline (https://github.com/pwilmart/PAW_pipeline) that both employ multiple testing correction. My concern is the existing list will contain so many false positives that it could compromise the value of the data to the field overall. This does not mean that the downstream enrichment analyses are compromised since you would expect the distribution of false hits to be more-or-less even across pathways and categories but the final list of DEPs in the publication should not be awash (or at least potentially so) in false positive hits.

Reviewer #2: (No Response)

7. PLOS authors have the option to publish the peer review history of their article (what does this mean?). If published, this will include your full peer review and any attached files.

Reviewer #1: No

Reviewer #2: No

---

## [Author Response · Author response to Decision Letter 1]

24 Sep 2020

Dear editors,

Thank you for your and reviewers’ valuable comments. We have updated the manuscript accordingly by addressing all points from you and the reviewers. Our revisions and response to the comments are listed as follows. All the changes are marked in the “Revised Manuscript with Track Changes”. 

Journal Requirements:

1. Comments: Please pay attention to the journal name in line 7 of cover letter. Please ensure this manuscript has not been published elsewhere and is not under consideration by another journal.

Responses: We confirm that this manuscript has not been published elsewhere and is not under consideration by another journal, and we have corrected it in our cover letter (line 7).

2. Comments: The results of TMT proteomics analysis can provide a new idea for the study of the pathogenesis of silicosis. However, the results of the manuscript only show the primary results of TMT analysis, and do not carry out in-depth data analysis, which is the biggest shortcoming of this manuscript. It is suggested that the authors make full use of the data to further explain the pathogenesis of silicosis. Considering that there are few reports on the proteomics of silicosis, the publication of this manuscript in Plos One is still of some significance.

Responses: As suggested, we have further explained the pathogenesis of silicosis mainly based on the proteins verified by PRM (line 300-392).

Reviewers' comments:

Reviewer #1:

Comments: While the textual changes and figure modifications have improved the overall quality of the manuscript. The authors have fundamentally failed to address my primary concern in their initial submission – namely false discoveries in their list of differentially expressed proteins (DEPs). In their response to reviewer comments, they highlight why they are confident in their peptide spectrum matches and the protein inferences are of high confidence. Those were not the issue. The question is how many of the proteins that pass the threshold of a simple t-test p<0.05 and a FC >1.8 are false positive DEPs. With >3000 tests, you would expect a large proportion of the proteins that passed a p<0.05 threshold to be random hits (~150). While there is some debate about how best to utilize multiple testing corrections (e.g. Benjamini and Hochberg) in small magnitude effect measurements like iTRAQ/TMT (https://onlinelibrary.wiley.com/doi/pdf/10.1002/pmic.201600044), it is something that should not just be ignored. If BH ends up being too aggressive, there are other options for at least characterizing what the likely FDR would be using a process like in DAPAR/Prostar (https://onlinelibrary.wiley.com/doi/pdf/10.1002/pmic.201600044) or additional filtering using a z-score (https://www.sciencedirect.com/science/article/pii/S187439191530186X?via%3Dihub). The authors could even run their data through existing analysis pipelines like MSstatsTMT(https://www.mcponline.org/content/early/2020/07/17/mcp.RA120.002105) or the PAW pipeline (https://github.com/pwilmart/PAW_pipeline) that both employ multiple testing correction. My concern is the existing list will contain so many false positives that it could compromise the value of the data to the field overall. This does not mean that the downstream enrichment analyses are compromised since you would expect the distribution of false hits to be more-or-less even across pathways and categories but the final list of DEPs in the publication should not be awash (or at least potentially so) in false positive hits.

Responses: We thank the reviewers for their constructive advices and helpful comments based on which we further defined differentially expressed proteins (DEPs) by using Benjamini and Hochberg False Discovery Rate (FDR) < 5% [ Pascovici D, et al., 2016, doi: doi:10.1002/pmic.201600044] (line 196-197). The result showed that 285 DEPs were all with FDR < 5%. Benjamini and Hochberg FDR-adjusted p-values of 285 DEPs were shown in S3 and S4 tables.

---

## [Decision Letter · Decision Letter 2]

13 Oct 2020

Comparative proteomic analysis of silica-induced pulmonary fibrosis in rats based on tandem mass tag (TMT) quantitation technology

PONE-D-20-07393R2

Dear Dr. Bo,

We’re pleased to inform you that your manuscript has been judged scientifically suitable for publication and will be formally accepted for publication once it meets all outstanding technical requirements.

Kind regards,

Yuqin Yao

Academic Editor

PLOS ONE

Additional Editor Comments (optional):

Reviewers' comments:

Reviewer's Responses to Questions

**Comments to the Author**

1. If the authors have adequately addressed your comments raised in a previous round of review and you feel that this manuscript is now acceptable for publication, you may indicate that here to bypass the “Comments to the Author” section, enter your conflict of interest statement in the “Confidential to Editor” section, and submit your "Accept" recommendation.

Reviewer #2: All comments have been addressed

2. Is the manuscript technically sound, and do the data support the conclusions?

Reviewer #2: Yes

3. Has the statistical analysis been performed appropriately and rigorously? 

Reviewer #2: Yes

4. Have the authors made all data underlying the findings in their manuscript fully available?

Reviewer #2: Yes

5. Is the manuscript presented in an intelligible fashion and written in standard English?

Reviewer #2: Yes

6. Review Comments to the Author

Reviewer #2: (No Response)

7. PLOS authors have the option to publish the peer review history of their article (what does this mean?). If published, this will include your full peer review and any attached files.

Reviewer #2: No

---

## [Editor Report · Acceptance letter]

19 Oct 2020

PONE-D-20-07393R2 

Comparative proteomic analysis of silica-induced pulmonary fibrosis in rats based on tandem mass tag (TMT) quantitation technology 

Dear Dr. Bo:

I'm pleased to inform you that your manuscript has been deemed suitable for publication in PLOS ONE. Congratulations! Your manuscript is now with our production department. 

Kind regards, 

on behalf of

Dr. Yuqin Yao 

Academic Editor

PLOS ONE